# Improving Conduct and Reporting of Narrative Synthesis of Quantitative Data (ICONS-Quant): protocol for a mixed methods study to develop a reporting guideline

Mhairi Campbell,[1] Srinivasa Vittal Katikireddi,[1] Amanda Sowden,[2] Joanne E McKenzie,[3] Hilary Thomson[1]

[1]MRC/CSO Social and Public Health Sciences Unit, University of Glasgow, Glasgow, UK
[2]Centre for Reviews and Dissemination, University of York, York, UK
[3]School of Public Health and Preventive Medicine, Monash University, Melbourne, Victoria, Australia

**Correspondence to**
Ms Mhairi Campbell;
Mhairi.Campbell@glasgow.ac.uk

## ABSTRACT

**Introduction** Reliable evidence syntheses, based on rigorous systematic reviews, provide essential support for evidence-informed clinical practice and health policy. Systematic reviews should use reproducible and transparent methods to draw conclusions from the available body of evidence. Narrative synthesis of quantitative data (NS) is a method commonly used in systematic reviews where it may not be appropriate, or possible, to meta-analyse estimates of intervention effects. A common criticism of NS is that it is opaque and subject to author interpretation, casting doubt on the trustworthiness of a review's conclusions. Despite published guidance funded by the UK's Economic and Social Research Council on the conduct of NS, recent work suggests that this guidance is rarely used and many review authors appear to be unclear about best practice. To improve the way that NS is conducted and reported, we are developing a reporting guideline for NS of quantitative data.

**Methods** We will assess how NS is implemented and reported in Cochrane systematic reviews and the findings will inform the creation of a Delphi consensus exercise by an expert panel. We will use this Delphi survey to develop a checklist for reporting standards for NS. This will be accompanied by supplementary guidance on the conduct and reporting of NS, as well as an online training resource.

**Ethics and dissemination** Ethical approval for the Delphi survey was obtained from the University of Glasgow in December 2017 (reference 400170060). Dissemination of the results of this study will be through peer-reviewed publications, and national and international conferences.

## INTRODUCTION

Well-conducted systematic reviews are important for informing clinical practice and health policy.[1] In some reviews, meta-analysis of effect estimates may not be possible or sensible. For example, data may be insufficient to allow calculation of standardised effect estimates, the effect metrics arising from different study designs may not be amenable to synthesis (eg, those arising from interrupted time series and randomised trials), or high levels of statistical heterogeneity may mean that presenting an average effect is misleading. For reviews of quantitative data where statistical synthesis is not possible, narrative synthesis of quantitative data (NS) is often the alternative method of choice. A major concern about NS is that it lacks transparency and therefore introduces bias into the synthesis.[2 3] This is an important criticism, which raises questions about the validity and utility of reviews using NS, and ultimately increases the risk of adding to research waste.[4] NS involves collating study findings into a coherent textual narrative, with descriptions of differences in characteristics of the studies including context and validity, often using tables and graphs to display results.[5 6] Published guidance for NS funded by the UK's Economic and Social Research Council (ESRC) describes techniques for promoting transparency between review level data and conclusions; these include graphical and structured tabulation

### Strengths and limitations of this study

► This study will be the first to develop a consensus-based reporting guideline for narrative synthesis of quantitative data (NS) in systematic reviews.
► The study follows the recommended methodology for developing reporting standards.
► The online Delphi survey of international experts in NS will be an effective method of gaining reliable consensus from a group of experts.
► The reporting guideline and the supplementary materials developed to support use of existing guidance will aid the implementation of best practice conduct and reporting of NS.

BMJ

of the data.[5] However, a recent analysis of systematic reviews of public health interventions suggests that this guidance is rarely used.[7]

Relative to developments in meta-analysis or statistical synthesis, and synthesis of qualitative data in the past decade, work to support improved conduct and transparent reporting in NS has been scarce. While a reporting guideline has been developed for systematic reviews and meta-analysis, the Preferred Reporting Items for Systematic Reviews and Meta-Analyses (PRISMA),[8] the focus of the synthesis items is on meta-analysis of effect estimates, with no items for alternative approaches to synthesis. The Cochrane Methodological Expectations of Cochrane Intervention Reviews (MECIR) standards for conducting and reporting Cochrane reviews specify one general item referring to non-quantitative synthesis or non-statistical synthesis, and do not have any items specifically for NS.[2] Reporting guidelines have had some impact on improving the reporting for randomised trials and may have similar benefits for improving the reporting of methods and results from NS.[9]

There is a growing demand for reviews addressing complex questions, and which incorporate diverse sources of data. Cochrane, a global leader in evidence synthesis of health and public health interventions, has recognised this.[10] Following the prioritisation of relevance and breadth of coverage in the Cochrane strategy,[10] it is likely that the proportion of Cochrane reviews addressing complex questions will increase; this may result in increased use of NS methods. Realising the need for improved implementation and reporting of NS methods, the Cochrane Strategic Methods Fund has funded the ICONS-Quant project: Improving Conduct and Reporting of Narrative Synthesis of Quantitative Data. This paper presents the protocol for the work that will be undertaken.

### ICONS-Quant

The ICONS-Quant project aims to improve the implementation of NS methods through enhancing existing guidance on the conduct of NS and developing a reporting guideline. Provision of reporting guidelines alone will not necessarily lead to improved research conduct; provision of explanatory guidance, dissemination, endorsement and support for adherence is also necessary.[11] We will produce materials to support the implementation of best practice in the application of NS methods, and improved reporting. While our focus is on Cochrane reviews, the key outputs of the project will be of use for reviews published elsewhere and will be made freely available. We will:

► describe current practice in conduct and reporting of NS in Cochrane reviews;
► achieve expert consensus on reporting standards for NS;
► provide support for those involved in NS through the provision of enhanced guidance on NS conduct and online training resources.

We intend ICONS-Quant guideline to be used in combination with the PRISMA guidelines.[8] The PRISMA guidelines provide items relating to the various stages of review conduct, for example, providing a clear abstract, explaining the literature search strategy, reporting methods to assess risk of bias. The ICONS-Quant reporting guideline will focus on the methods of synthesis, relating most closely to expanding on PRISMA Item 14 'synthesis of results', outlining details that require to be reported to promote transparency in NS.

## METHODS AND ANALYSIS

The ICONS-Quant project will be conducted over a period of 24 months from May 2017. Here, we outline the development of a reporting guideline for NS and supporting materials for existing guidance. In line with recommendations for best practice in developing reporting guidelines,[11] we will:

► identify the need for the ICONS-Quant guideline (Work Programme One);
► conduct a Delphi survey and consensus meeting (Work Programme Two);
► enhance existing guidance on NS (Work Programme Three);
► develop learning materials for implementation of NS (Work Programme Four).

Below we outline the Project Advisory Group (PAG) and the research that will be conducted within each Work Programme. Details of the ICONS-Quant project have been registered with the Enhancing the Quality and Transparency of Health Research Network, which provides a database of reporting guidelines in development (http://www.equator-network.org/library/reporting-guidelines-under-development/#74).

### Project Advisory Group

We have established an ICONS-Quant PAG which will provide governance for the project as well as expert advice. The ICONS PAG includes named project collaborators from Cochrane Review Groups (Effective Practice and Organisation of Care, Consumers and Communication, and Tobacco Addiction), a representative with experience of NS from the Campbell Collaboration Methods Group and a user representative from the National Institute for Health and Care Excellence.

### Work Programme One: assessment of current reporting and conduct of NS in Cochrane reviews

Previously we investigated current practice in the conduct and reporting of NS in systematic reviews of public health interventions.[7] Work Programme One will extend this exercise to assess use of NS methods and their reporting across all Cochrane Review Groups. We will identify all Cochrane reviews published between April 2016 and April 2017 and screen them to determine the method of synthesis for the primary outcome. Reviews will be included for further examination if the method for

reporting the synthesis of the primary outcomes relies on text. We will identify those that use NS or that synthesise studies using text only, whether or not the authors refer to the use of NS or textual methods for synthesis. Reviews will be excluded if they are empty, include only one study, report on diagnostic test accuracy, or are a review of methodology. We will record how the synthesis has been conducted and reported. We will use the existing data extraction template designed for our previous assessment of NS in public health reviews. This template is based on key sources of best practice for NS,[12–15] including the ESRC guidance on the conduct of NS.[5] Questions relate to use of theory; investigation of differences across included studies and reported findings; transparency of links between data and text (including data visualisation tools used); assessment of robustness of the synthesis; and adequacy of description of NS methods.[5] Using a similar format to our review of NS in public health reviews,[16] we will tabulate the extracted data. This will allow description of:

► the extent of reporting of NS methods: the amount and type of detail included;
► the range of approaches and tools used to narratively synthesise data;
► how conceptual and methodological heterogeneity is managed;
► review authors' reflection on robustness of synthesis.

The results of this exercise will be used to inform development of the initial checklist for inclusion in the Delphi survey.

### Work Programme Two: Delphi survey

A Delphi consensus survey will be conducted. This is the standard approach to elicit expert opinion for the purposes of developing consensus-based reporting guidelines.[17 18] The results of the assessment exercise in Work Programme One, in conjunction with key texts on NS,[12–15 19 20] findings from the previous assessment of reporting NS in public health reviews[16] and input from the ICONS PAG, will be used to develop the initial items for Round One of the Delphi survey. An expert panel will then be consulted to inform the development of the Delphi survey. The panel will be identified by the project team and members of the ICONS PAG, and will comprise 15–20 authors and methodologists experienced in or familiar with the purpose and conduct of NS. A videoconference with the expert panel will be used to present findings from Work Programme One and a draft of the proposed Delphi survey. Participants' input will be recorded and used to refine the Delphi survey.

The Delphi online survey will use a questionnaire to achieve consensus on the content and wording of reporting items considered to capture the pertinent details of NS. The online platform will be created by the MRC/CSO Social and Public Health Sciences Unit, University of Glasgow, using a web-based platform recently developed for this purpose. The platform facilitates personalised invitations to participate, password-protected logins and personalised reminders, and enables data collation for quantitative and qualitative analyses. There will be two rounds of the survey, with a third version conducted if necessary to gain consensus among participants.

Participants will include members of the ICONS PAG and others experienced in NS. Suitable participants will be identified by the project team, through recommendations from the ICONS PAG and through the data extraction exercise described in Work Programme One. The data extraction exercise will help articulate identified gaps in reporting of methods and findings of NS where transparency is particularly lacking. The identified gaps will be used when drafting reporting item questions for the Delphi consensus exercise, to improve transparency in NS. We will invite a maximum of 100 individuals to participate and they will be recruited via their workplace email address. We will ask for their professional opinion on the content of a draft reporting guideline. The invitation will outline the aim of the Delphi survey, the process involved and the time commitment, and include a participant information sheet. Individuals who accept the invitation will be asked to take part in each round of the survey. It will be clearly stated that at any stage, a respondent can opt out of the Delphi survey. The survey will ask participants to provide details of their job category; no personal information will be collected. Respondents will be asked to use their email address to log in to the survey. This information will be used only to verify the appropriate use of the survey and will not be used in the analysis. The Delphi survey will involve implied consent: it will be made clear to participants that by responding to the survey, they are consenting to participate in the study. It will be explained to respondents that: their responses will not be linked to their identity (deidentified); only researchers will have access to the data; and the data will be stored on a password-encrypted computer and stored and destroyed in accordance with Medical Research Council guidelines.

### Round One

The Delphi survey will consist of closed and open-ended questions. Round One of the survey will provide an introduction to the project and instructions for the survey. The participants will be invited to rank each of the proposed guideline items on a 4-point Likert scale (essential, desirable, possible, omit, used in previous Delphi surveys for developing reporting guidelines[21 22]). For each item, the participants will be invited to provide comments. A reminder email will be sent approximately 2 weeks after the initial invitation. Round One will close approximately 4 weeks after the first invitations are issued. Responses to Round One of the Delphi will be exported verbatim into a Microsoft Excel spreadsheet and collated. Responses to the scale rating will be summarised as counts and percentage frequencies. The free-text content will be collated and summarised. The results from both the quantitative and qualitative data collation will be used to inform the development of Round Two of the Delphi and the content of the final guideline checklist. Redrafting of

the Delphi survey items will be conducted in discussion with all study group members within 1 month of closure of the round.

## Round Two

All participants from Round One will be invited to take part in Round Two. In Round Two, the proposed checklist items will be presented in three sections:

1. Items that reached high consensus in Round One and that are expected to be included in the final checklist. These items will have an a priori agreement of >70% approval, as recommended by Diamond *et al*.[23] Participants will not be asked to rate these items again but will be asked to comment on whether they agree with the inclusion of each item in the checklist and to provide comments if they disagree or with suggestions to clarify the wording of the items.
2. Items that have been significantly altered or are additional as a result of Round One. The participants will be invited to rate these items on the 4-point scale and provide comments on each item.
3. Items that were rated as 'omit' in Round One and that are not expected to be included in the final checklist. Participants will not be asked to rate these items; they will be invited to provide their opinion on the removal of these items from the final checklist.

If there is a substantial lack of consensus remaining following Round Two of the Delphi, a third round will be prepared and conducted. Round Three will follow the same format as Round Two, providing the reporting guideline items in three sections: items that are expected to be included in the final guideline; those significantly altered; and items that will be removed from the final checklist.

## Consensus meeting

An expert panel of individuals experienced in NS methods will be invited to participate in the consensus meeting to finalise the content of the guideline. It is anticipated that this will be held as a face-to-face meeting at the Cochrane colloquium in 2018 in Edinburgh, UK. If this is not possible, an online consensus meeting will be conducted using webinar software. If necessary, an additional virtual meeting will be held to accommodate different time zones of invitees. At the consensus meeting the reporting guideline items developed from the Delphi survey will be discussed, with priority given to establishing consensus on the content and wording of items for which the level of consensus is less clear.

## Work Programme Three: enhancement of existing guidance on NS methods

We will produce materials to support the current guidance that includes information on the rationale for, as well as implementation of each stage of NS. This will be developed as a supplement to the reporting guideline items. The enhanced guidance will be accessible to novice reviewers and will provide examples of good practice to

illustrate how methods of NS may be used. The findings of Work Programmes One and Two, the assessment of current reporting of NS and the Delphi consensus will be used to inform development of enhanced guidance on NS.[5] Cochrane Review Groups who publish reviews incorporating NS will be identified through the process of Work Programme One. We anticipate that these will include a range of Cochrane Review Groups and examples will be developed which are relevant to all groups. An overview of methodological tools which can be used to support NS and which have been developed since publication of the ESRC guidance in 2006 will also be incorporated. The PAG will be asked for comments on the draft guidance before it is piloted.

## Work Programme Four: development of learning materials on implementation of NS

Training materials based on the guidance developed in Work Programmes Two and Three will be produced to promote improved use of NS methods. We have secured support from Cochrane Training to collaborate in Work Programme Four. We will deliver two to three live participatory webinars (to allow for different time zones) to present the agreed guidance developed in Work Programme Two. One webinar will be recorded and provided on a web page, along with a record of the questions raised in the webinar, and any other frequently asked questions that emerge.

In addition, an online training module on NS will be developed in collaboration with Cochrane Training and a specialist e-learning company. The module will include a mix of didactic and participatory teaching methods involving assessment and interpretation of data and syntheses. We will work with Cochrane colleagues to incorporate the reporting items into the MECIR standards, and offer to update the relevant chapters of the Cochrane Handbook.

## ETHICS AND DISSEMINATION

Dissemination of the results of this study will be through peer-reviewed publications, and national and international conferences. In addition, the objectives of Work Programmes Three and Four are to distribute and encourage use of the ICONS-Quant guideline through webinars and an online training module.

**Contributors** HT conceived the idea of the study. HT, SVK, AS, JEM and MC designed the study methodology. MC prepared the first draft of the protocol manuscript and all authors critically reviewed and approved the final manuscript.

**Funding** This project was supported by funds provided by the Cochrane Strategic Methods Fund. MC, HT and SVK receive funding from the UK Medical Research Council (MC_UU_12017-13 and MC_UU_12017-15) and the Scottish Government Chief Scientist Office (SPHSU13 and SPHSU15). SVK is supported by an NHS Research Scotland Senior Clinical Fellowship (SCAF/15/02). JEM is supported by a National Health and Medical Research Council (NHMRC) Australian Public Health Fellowship (1072366).

**Disclaimer** The views expressed in the protocol are those of the authors and not necessarily those of Cochrane or its registered entities, committees or working groups.

**Competing interests** HT and SVK are Cochrane editors. JEM is a co-convenor of the Cochrane Statistical Methods Group.

**Patient consent** Not required.

**Ethics approval** University of Glasgow College of Social Sciences Ethics Committee (reference number400170060)

**Provenance and peer review** Not commissioned; externally peer reviewed.

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
