## [Reviewer comments · BMJ Open]

ARTICLE DETAILS

TITLE (PROVISIONAL)	Improving conduct and reporting of narrative synthesis of quantitative data (ICONS-Quant): protocol for a mixed methods study to develop a reporting guideline
AUTHORS	Campbell, Mhairi Katikireddi, Srinivasa Sowden, Amanda McKenzie, Joanne Thomson, Hilary

VERSION 1 – REVIEW

REVIEWER	G.J. Melendez-Torres DECIPHer, Cardiff University, UK
REVIEW RETURNED	01-Nov-2017

GENERAL COMMENTS	I have very few questions about this protocol. It represents work that is timely, necessary and rigorous. --In WP1, do you expect to run into any issues with the depth of reporting? It has been our group's experience that reporting of how the narrative synthesis was undertaken is generally scant. Do you think you might make up with relatively impoverished reporting within studies by examining across studies? --How will you articulate this work in relation to existing reporting guidelines for systematic reviews of interventions, specifically PRISMA and RAMESES? --Can you provide more detail on how work from WP1 will inform the questions for the Delphi survey?
--

REVIEWER	Andrew Booth School of Health and Related Research (ScHARR), University of Sheffield
REVIEW RETURNED	05-Nov-2017

GENERAL COMMENTS	This is a well constructed and well-written protocol following the methodology recommended for reporting standards by Moher and the EQUATOR group. The work is strategically important as seen by the funding from Cochrane and the case made for poor uptake of the ESRC document. The science is sound and the authors seem to have anticipated all major considerations for a project of this type. As someone who includes narrative synthesis in my teaching I look
---

	forward to seeing the results of this work. ----- Introduction: “For example, sufficient data may not allow calculation of standardised effect estimates,” This seems a perverse way of phrasing this important point which clouds meaning. I think the authors mean – “For example, data may be insufficient to allow....” “and ultimately adding to research waste” – This shorthand makes meaning unclear. Spell out what is meant i.e. “and ultimately increases the risk of adding to research waste”. ----- “The Cochrane Methodological Expectations of Cochrane Intervention Reviews (MECIR) provide standards for conducting and reporting Cochrane reviews. They include one general item referring to non-quantitative synthesis or non-statistical synthesis, and do not have any items specifically for NS.[2]” These sentences do not flow and would be better expressed as: “The Cochrane Methodological Expectations of Cochrane Intervention Reviews (MECIR) standards for conducting and reporting Cochrane reviews specify one general item referring to non-quantitative synthesis or non-statistical synthesis, and do not include any items specifically for NS.[2]” ----- “We will identify Cochrane reviews published between April 2016 and April 2017 that use NS or that synthesise studies using text only, and record how the synthesis has been conducted and reported.” It is not clear how the authors will identify these items. (1) Will they examine the sample of all Cochrane reviews published during this time and include all those that refer explicitly to narrative synthesis; (2) Will they examine all Cochrane reviews during this time and include all those that cite the ESRC work; (3) Will they include all Cochrane reviews during this time that use narrative synthesis whether doing this explicitly or not (4) will they be retrieving the included reviews by a search strategy or by sifting? A brief clarification would be helpful here. “(essential, desirable, possible, omit).” Have these categories been used previously? I am not sure that “desirable” and “possible” are sufficiently mutually exclusive – they seem to relate to two different domains ie acceptability and feasibility? If these are accepted and workable categories then some reference to their prior use would strengthen confidence in the Methods.
--	---

VERSION 1 – AUTHOR RESPONSE

We thank the editor and reviewers for their comments which have helped us clarify and improve our protocol.

Editor Comments to Author:

- When will ethics approval be obtained? Please update the paper.

RESPONSE: A submission has been made to the University of Glasgow College of Social Sciences Ethics Committee in November 2017. The manuscript has been updated to state this.

Edited text, page 2, paragraph 3: "Application for ethics approval for the Delphi survey was submitted to the University of Glasgow in November 2017 (reference 400170060)."

Edited text, page 12, paragraph 3: "An application for ethics approval for the Delphi survey was submitted to the University of Glasgow College of Social Science Ethics Committee in November 2017 (reference number 400170060)."

- The Strengths and Limitations section should just consist of points about the strengths and limitations of the study and study design. It should not provide any summary points.

RESPONSE: The summary points of the project have been removed from the Strengths and Limitations section. The revised summary points are:

- This study will be the first to develop a consensus based reporting guideline for narrative synthesis of quantitative data in systematic reviews.
- The study follows the recommended methodology for developing reporting standards.
- The online Delphi survey of international experts in NS will be an effective method of gaining reliable consensus from a group of experts.
- The reporting guideline and the supplementary materials developed to support use of existing guidance will aid the implementation of best practice conduct and reporting of NS.

Reviewer(s)' Comments to Author:

Reviewer: 1

Reviewer Name: G.J. Melendez-Torres

--In WP1, do you expect to run into any issues with the depth of reporting? It has been our group's experience that reporting of how the narrative synthesis was undertaken is generally scant. Do you think you might make up with relatively impoverished reporting within studies by examining across studies?

RESPONSE: We are unsure what this question is asking, if the reviewer is asking whether we want to consider conducting more in-depth analysis of the reviews in our sample report, our response is: The main purpose of WP1 is to establish what is currently being done in narrative synthesis in Cochrane reviews. If we find lack of depth in the reporting of narrative synthesis methods, this will be an important finding of the work package. While it would be interesting to examine across reviews, we believe the current work package fits the intended purpose to describe what methods are used and reported in Cochrane reviews using narrative to synthesise the data. The data extraction template we have developed will extract information on how the methods are reported and how the reviews are conducted. Therefore, even if there is no report of the method, we will assess the key components of narrative synthesis by how the findings of the synthesis are reported and interpreted. For example, we will examine how heterogeneity is managed by looking at how any sub-groups of data or included studies were organised, and whether visual presentation of the included data results reflected the groupings used in the narrative. If required, the full data extraction template could be provided as an online supplementary file.

--How will you articulate this work in relation to existing reporting guidelines for systematic reviews of interventions, specifically PRISMA and RAMESES?

RESPONSE: Prior to securing funding for this project, we contacted the PRISMA developers to discuss this reporting guideline being a supplement to PRISMA. The response was that reporting guidelines on narrative synthesis were beyond the PRISMA scope which focusses on reviews which rely on meta-analysis. The RAMESES guideline is not directly relevant as it focusses on how to report realist synthesis as an alternative approach to systematic review. In addition, the PRISMA and the RAMESES guidelines incorporate reporting items for the complete review process, for example searching etc., the ICONS- Quant reporting guideline will focus on the methods of synthesis, relating most closely to PRISMA Item 14. The final ICONS reporting tool will make this clear to users of the tool, pointing out that reporting standards for other aspects of review conduct are detailed in PRISMA. We have updated the text to explain that ICONS extends PRISMA item 14.

Edited text, page 6, paragraph 2: "We intend ICONS-Quant guideline to be used in combination with the PRISMA guidelines.[8] The PRISMA guidelines provide items relating to the various stages of review conduct, e.g. providing a clear abstract, explaining the literature search strategy, reporting methods to assess risk of bias. The ICONS-Quant reporting guideline will focus on the methods of synthesis, relating most closely to expanding on PRISMA Item 14 'synthesis of results', outlining details that require to be reported to promote transparency in NS".

--Can you provide more detail on how work from WP1 will inform the questions for the Delphi survey?

RESPONSE: The focus of WP1 is to explore the extent to which recent Cochrane reviews report narrative synthesis, according to the most recent guidance available on best practice in narrative synthesis of quantitative data (including ESRC guidance by Popay et al 2006). The data extraction in WP1 aims to establish current practice in reporting of narrative synthesis of quantitative data. This includes both reporting of methods used, and reporting of the synthesis findings, including related data and conclusions of the synthesis. The data gathered from this exercise will be used to help articulate identified gaps in reporting of methods and findings of narrative synthesis where transparency is particular lacking. We believe this will help us develop clear guideline items relevant to current practice of narrative synthesis of quantitative data.

The text has been updated to make this clearer, edited text, page 9, paragraph 1: "The data extraction exercise will help articulate identified gaps in reporting of methods and findings of narrative synthesis where transparency is particular lacking. The identified gaps will be used when drafting reporting item questions for the Delphi consensus exercise, to improve transparency in NS."

Reviewer: 2

Reviewer Name: Andrew Booth

Institution and Country: School of Health and Related Research (ScHARR), University of Sheffield

Please state any competing interests or state 'None declared': "None declared"

Please leave your comments for the authors below This is a well constructed and well-written protocol following the methodology recommended for reporting standards by Moher and the EQUATOR group. The work is strategically important as seen by the funding from Cochrane and the case made for poor uptake of the ESRC document. The science is sound and the authors seem to have anticipated all major considerations for a project of this type. As someone who includes narrative synthesis in my teaching I look forward to seeing the results of this work.

Introduction: "For example, sufficient data may not allow calculation of standardised effect estimates," This seems a perverse way of phrasing this important point which clouds meaning. I think the authors mean – "For example, data may be insufficient to allow...."

RESPONSE: Thank you for the simplified phrasing, the text has been revised.

“and ultimately adding to research waste” – This shorthand makes meaning unclear. Spell out what is meant i.e. “and ultimately increases the risk of adding to research waste”.

RESPONSE: Thank you for the simplified phrasing, the text has been revised.

“The Cochrane Methodological Expectations of Cochrane Intervention Reviews (MECIR) provide standards for conducting and reporting Cochrane reviews. They include one general item referring to non-quantitative synthesis or non-statistical synthesis, and do not have any items specifically for NS.[2]” These sentences do not flow and would be better expressed as:

“The Cochrane Methodological Expectations of Cochrane Intervention Reviews (MECIR) standards for conducting and reporting Cochrane reviews specify one general item referring to non-quantitative synthesis or non-statistical synthesis, and do not include any items specifically for NS.[2]”

RESPONSE: Thank you for the simplified phrasing, the text has been revised.

“We will identify Cochrane reviews published between April 2016 and April 2017 that use NS or that synthesise studies using text only, and record how the synthesis has been conducted and reported.”

It is not clear how the authors will identify these items. (1) Will they examine the sample of all Cochrane reviews published during this time and include all those that refer explicitly to narrative synthesis;

(2) Will they examine all Cochrane reviews during this time and include all those that cite the ESRC work; (3) Will they include all Cochrane reviews during this time that use narrative synthesis whether doing this explicitly or not (4) will they be retrieving the included reviews by a search strategy or by sifting? A brief clarification would be helpful here.

RESPONSE: (1) All Cochrane reviews published during this time will be identified. These will be screened to establish whether the primary outcome(s) are synthesised by meta-analysis or by using text. Our focus is on reviews that are using methods other than meta-analysis, as the main way synthesis other than meta-analysis is described is by using narrative. In response to query (1) we will not limit our screening to only include reviews that use the phrase ‘narrative synthesis’. (2) we will not limit screening to only include reviews that cite the ESRC guidance. (3) yes, we will include all reviews that use narrative synthesis, whether or not the review authors are explicit about this. The text has been updated to clarify the inclusion criteria, see below.

(4) The simple search strategy is to search the Cochrane database for all reviews published between April 2016 and April 2017 and screen all of these to determine the method of synthesis for the primary outcome(s). Reviews will be included if the method for reporting the synthesis relies on text. Reviews will be excluded if they are empty, include only one study, report on diagnostic test accuracy, or are a review of methodology.

The text has been updated to clarify the inclusion criteria, edited text page 7, paragraph 2: “We will identify all Cochrane reviews published between April 2016 and April 2017 and screen them to determine the method of synthesis for the primary outcome. Reviews will be included for further examination if the method for reporting the synthesis of the primary outcomes relies on text. We will

identify those that use NS or that synthesise studies using text only, whether or not the authors refer to the use of narrative synthesis or textual methods for synthesis. Reviews will be excluded if they are empty, include only one study, report on diagnostic test accuracy, or are a review of methodology. We will record how the synthesis has been conducted and reported.”

“(essential, desirable, possible, omit).” Have these categories been used previously? I am not sure that “desirable” and “possible” are sufficiently mutually exclusive – they seem to relate to two different domains ie acceptability and feasibility? If these are accepted and workable categories then some reference to their prior use would strengthen confidence in the Methods.

RESPONSE: The four categories have been used previously when developing the guideline for reporting interventions (TIDieR) by Hoffmann and colleagues, and again more recently for developing a guideline for reporting population health and policy interventions (TIDieR-PHP) by two of the authors of this protocol. On both occasions, the categories worked well with Delphi participants. The text has been updated to refer to the previous use of the categories.

Edited text, page 9 paragraph 2: “The participants will be invited to rank each of the proposed guideline items on a four-point Likert Scale (essential, desirable, possible, omit, used in previous Delphi surveys for developing reporting guidelines[21,22]).

VERSION 2 – REVIEW

REVIEWER	G.J. Melendez-Torres DECIPHer, Cardiff University, UK
REVIEW RETURNED	28-Nov-2017
GENERAL COMMENTS	Authors have satisfactorily attended to all points raised.